# Using In Silico Approach for Metabolomic and Toxicity Prediction of Alternariol

**DOI:** 10.3390/toxins15070421

**Published:** 2023-06-28

**Authors:** Daniela Eliza Marin, Ionelia Taranu

**Affiliations:** National Research and Development Institute for Biology and Animal Nutrition, 077015 Balotesti, Romania; daniela.marin@ibna.ro

**Keywords:** alternariol, metabolism, toxicity

## Abstract

Alternariol is a metabolite produced by *Alternaria* fungus that can contaminate a variety of food and feed materials. The objective of the present paper was to provide a prediction of Phase I and II metabolites of alternariol and a detailed ADME/Tox profile for alternariol and its metabolites using an in silico working model based on the MetaTox, SwissADME, pKCMS, and PASS online computational programs. A number of 12 metabolites were identified as corresponding to the metabolomic profile of alternariol. ADME profile for AOH and predicted metabolites indicated a moderate or high intestinal absorption probability but a low probability to penetrate the blood–brain barrier. In addition to cytotoxic, mutagenic, carcinogenic, and endocrine disruptor effects, the computational model has predicted other toxicological endpoints for the analyzed compounds, such as vascular toxicity, haemato-toxicity, diarrhea, and nephrotoxicity. AOH and its metabolites have been predicted to act as a substrate for different isoforms of phase I and II drug-metabolizing enzymes and to interact with the response to oxidative stress. In conclusion, in silico methods can represent a viable alternative to in vitro and in vivo tests for the prediction of mycotoxins metabolism and toxicity.

## 1. Introduction

*Alternaria* molds can contaminate a wide variety of food and feed materials, such as cereals, oilseeds, apples, tomatoes, olives, and several other fruits and vegetables [1]. These fungi produce several toxins, the most important being alternariol (AOH), alternariol monomethyl ether (AME), tenuazonic acid, altenuene, and altertoxins [2].

*Alternaria* mycotoxins can lead to significant economic losses and negatively impact human and animal health [3]. Among *Alternaria* mycotoxins, the toxicity of AOH and AME, the toxins responsible for genotoxic, mutagenic, cytotoxic, and carcinogenic effects, has been investigated the most [4]. These toxins act as topoisomerase poisons, generating DNA strand breaks and gene mutations in cultured human and animal cells by altering the redox balance [4].

*Alternaria* toxins, including AOH, belong to the group of “emerging” mycotoxins that are not routinely determined and not regulated by legislation but for which there is increasing evidence of their occurrence and toxic effects [5]. AOH has been found in unprocessed cereals and cereal products from Europe (n = 1665) in concentrations ranging between 0.75 and 832 mg/kg [2], but also in other commodities as fruits, legumes, nuts, oils, and oilseeds [1,6,7]. There are no regulations at present concerning the amount of AOH concentration in food and feed. This is despite the fact that, according to the EFSA, human dietary exposure to alternariol exceeds the threshold of toxicological concern, and there is increasing evidence concerning the toxic effect of AOH [8]. The EFSA considers that studies assessing the toxicological properties of *Alternaria* mycotoxins are insufficient for a correct health risk assessment to be made, thus making the establishment of specific regulations impossible [9].

Living organisms have biological systems that facilitate the elimination of xenobiotics, including mycotoxins [10]. In plants, metabolic transformations of mycotoxins generate modified forms of mycotoxins called “masked mycotoxins”, which are mainly metabolites that have resulted from phase-II plant metabolism [11]. These metabolites can co-contaminate food and feed along with their parent compounds, which can result in an underestimation of the actual amount of mycotoxins in foods [12,13]. Some papers reported the simultaneous presence of *Alternaria* toxins and their metabolites (AOH-3-sulfate, AOH-9-glucoside, AOH-3-glucoside, AME-3-sulfate) in samples of different origins [14]. In animals, the biotransformation of drugs and xenobiotics is realized mainly through oxidative (Phase I) or conjugative (Phase II) reactions. In some situations, metabolization of mycotoxins can lead to compounds with higher toxicity than the parent compounds, such as the transformation of aflatoxin B1 into AFB1 epoxide by cytochrome P450 enzymes that are responsible for DNA damage in the liver [15]. Due to a limited number of in vivo experiments, data concerning *Alternaria* toxins metabolism in animal organisms are scarce, but the prediction of metabolism products is important for toxicological studies in order to identify their effects in humans and animal organisms. Few data concerning the toxicological effects of AOH metabolites are available in the literature. For example, the DNA strand-breaking potential of AOH decreases significantly following hydroxylation or glucuronidation reactions [16,17]. Additionally, AOH and its metabolite, AOH-3-O-sulfate, may have a similar interaction with estrogen receptors, as described in an in silico study, indicating a possible comparative estrogenic effect of the two compounds [18]. For identifying harmful toxins, computational methods represent a viable alternative to in vivo animal tests that are expensive, time consuming, raise ethical considerations, and should be limited according to the principle of 3Rs. In silico toxicology aims to generate rapid results for toxicity prediction that can be complementary to the existing toxicity data, providing a general frame for future toxicity tests and avoiding possible failure in their design [19]. Lately, there have been important advances regarding the in silico modeling of absorption, distribution, metabolism, and excretion (ADME) properties for the assessment and prediction of toxicological effects that have resulted in a large number of free commercial in silico prediction tools [20,21]. Previous studies have used in silico approaches in order to assess the estrogenic potential of *Alternaria* mycotoxins and other xenoestrogens. A recent study by Dellafiora et al., has shown that in silico approaches can be used as useful tools for assessing differences between species in terms of mycotoxin estrogenicity [22]. Additionally, an in silico approach was used for assessing the xenoestrogenic potential of *Alternaria* mycotoxins and metabolites, indicating that methylation reaction can increase AOH estrogenicity [18].

The objective of the present paper was to provide (i) a prediction of Phase I and II metabolites of alternariol; and (ii) a detailed ADME/Tox profile for alternariol and predicted metabolites using an in silico working model based on the following computational programs: MetaTox, SwissADME, pKCMS, and PASS online.

## 2. Results

### 2.1. Metabolomics Profile of Alternariol

The prediction of Phase I and II metabolites of alternariol were realized by MetaTox software based on the calculation of the probability of their formation, starting from the AOH canonical SMILE structure. Twelve compounds were proposed, as predicted metabolite products form Phase I and II reactions.

Figure 1 illustrates the metabolomic profile of alternariol, predicted by MetaTox software, resulting from the reaction of aromatic hydroxylation (three predicted metabolites: M1–M3), which corresponds to Phase I metabolites, or from the reaction of o-glucuronidation (three predicted metabolites: M4–M6), o-sulfation (three predicted metabolites: M7–M9), and methylation (three predicted metabolites: M10–M12), which correspond to Phase II metabolites.

### 2.2. Physicochemical Properties, Pharmacokinetic Predictions and Drug Likeness

SwissADME and pharmacokinetic servers pKCMS were used in order to predict and calculate physicochemical properties, drug likeness, and the ADME/Tox-related descriptors of AOH and AOH-predicted metabolites. These two web servers were selected based on the accessibility (free access) and robustness of their computational methods used for the estimation of the pharmacokinetics and toxicity of small molecules [23,24]. Additionally, these methods have been extensively used and validated with experimental data [25,26,27]. The 2D chemical structure of AOH and metabolites was imported from MetaTox in SwissADME molecular sketcher, based on ChemAxon’s Marvin JS, in order to obtain the canonical SMILE for each compound (Table 1) and to predict the physicochemical properties, pharmacokinetic predictions, and drug likeness (Table 2). Drug likeness represents the probability that a molecule can be an oral active drug based on its bioavailability and was assessed via SwissADME, which filters chemical libraries in order to exclude incompatible molecules [23].

The Lipinski filter characterizes small molecules based on their physicochemical property profiles, which include a molecular weight (MW) less than 500, lipophilicity (cLogP) < 5, hydrogen bond acceptor (HBA) ≤ 10, and hydrogen bond donor (HBD) ≤ 5 [28]. 

Out of the thirteen mycotoxins listed in Table 2, ten of them do not violate Lipinski’s rule of five for oral availability, except for metabolites M4–M6, which resulted from the o-glucuronidation reaction, suggesting a low permeability or poor absorption for these compounds [28]. 

The probability of ADMET for AOH and predicted metabolites was evaluated using pkCSM software. As shown in Table 3, permeability coefficients across monolayers of Caco-2 (human colon carcinoma cell line) used for the prediction of the absorption of orally administered drugs show that, with the exception of metabolites resulting from the reaction of O-glucuronidation (M4–M6), all other mycotoxins have coefficients between 0.388 and 1.057; only AOH and M10 have coefficients of Caco-2 permeability superior to 1. Correlated with the results concerning drug likeness for the M4–M6 metabolites, intestinal absorption has very low values (between 14.9–18.04%). The intestinal absorption mildly increases for the metabolites resulting from the o-sulfation reaction (45–48.6%) and from aromatic hydroxylation (73.6–81.8%), while reaching a very high percentage of absorption for AOH (95.4%) and the metabolites resulting from the methylation reaction (95.6–97.08%). The skin permeability (Kp) expressed as permeation coefficients (logKp) refers to the rate of drug penetrating across the stratum corneum, and the predicted values were similar for all compounds with a value of (−2.7). P-glycoprotein plays a significant role in drug absorption and distribution by limiting the cellular uptake of drugs from blood circulation into the brain and from the intestinal lumen into epithelial cells [29]. The volume of distribution (VDss) is an in vivo pharmacokinetic parameter that provides an indication of the probability of drug distribution in the body [30]. AOH and metabolites were predicted to be substrates for P-glycoprotein, while they could not act as inhibitors for P-glycoprotein I or II. AOH metabolites resulting from aromatic hydroxylation (M1–M3) or methylation (M10–M11) have high VDss values, suggesting their significant concentration in tissues due to, for example, high lipid solubility or tissue binding. On the other side, low VDss values were predicted for metabolites resulting from o-glucuronidation and o-sulfation, indicating rather high plasma protein binding or high water solubility [31].

Unbound fractions, also known as “free” forms, are responsible for toxic effects because they can act on tissue target sites. Unbound fraction values varied between 0.083 and 0.22, with the lowest values for the metabolites of aromatic hydroxylation (0.083–0.112) and the highest for M4–M6 (0.193–0.22) and M9 (0.22). The results presented in Table 2 indicate a rather low probability that the analyzed mycotoxins penetrate the brain–blood barrier or central nervous system. Metabolism prediction through pkCSM showed no probability that AOH and predicted metabolites could be substrates for CYP2D6 and CYP3A4. Metabolites from o-glucuronidation were not predicted as inhibitors of cytochromes CYP1A2, CYP2C19, and CYP2C9. None of the analyzed compounds inhibit the cytochromes CYP2D6 and CYP3A4. AOH, metabolites from aromatic hydroxylation and methylation, as well as M7 and M8, inhibit CYP1A2, while M1, M2, M11, and M12 may inhibit CYP2C9 (Table 2). Concerning the excretion, total drug clearance went from 0.65 to 0.84 log mL/min/kg, while none of the metabolites were predicted as renal OCT2 substrates (Table 2).

### 2.3. Prediction of Toxicity

Pharmacokinetic servers PASS online and pKCMS were used in order to predict AOH and metabolite toxicity. Only metabolites M3 and M11 were predicted to have potential genotoxicity as a result of the AMES toxicity index, representing the compound’s ability to induce reverse mutations at the selected loci of several bacterial strains (Table 4).

Analysis of the tolerated dose, equivalent to the highest mycotoxin dose that does not cause unacceptable side effects, was predicted to have a higher value for the metabolites resulting from aromatic hydroxylation (from 0.83 to 0.94) and o-sulfation (from 0.91–0.97). Metabolites resulting from methylation (M10–M12) have the highest oral acute toxicity in rats, calculated as LD50 (lethal dose 50), while the metabolites resulting from o-glucuronidations (M4–M6) seem to be more involved in chronic toxicity in rats after oral exposure, expressed as LOAEL (lowest observed adverse effect level). Additionally, M4–M6 metabolites were predicted to be the most toxic in minnows, while the toxicity in *Tetrahymena piriformis* looks to be similar for all the analyzed compounds, with values ranging from 0.285 to 0.349.

The toxicological endpoints were predicted using PASS on-line software, which predicts more than 4000 types of biological activity, including toxic effects, mechanisms of action, interaction with transporters and metabolic enzymes, and influence on gene expression [32]. The results for AOH and the metabolites are shown in Figure 2. As shown on the heatmap, AOH toxicity seems to be lower than that of its metabolites, with the exception of the metabolites resulting from the methylation reactions (M10–M12). The metabolite products of aromatic hydroxylation have similar toxicological endpoints to the parent compound, while the metabolites resulting from glucuronidation and sulfation show a wide and different toxicity profile (Figure 2).

With the exception of M9, all the mycotoxins induce vascular toxicity. Mycotoxins resulting from glucuronidation and sulfation were predicted to have carcinogenic, teratogen, and embryotoxic potential, while only M4–M6 were able to interact with inflammation, have a nephrotoxic effect, and have the potential to be an endocrine disruptor.

The prediction effects of parent mycotoxins and predicted metabolites on different isoforms of Phase I enzymes (cytochrome P450) involved in the xenobiotic metabolization are presented in Figure 3.

Effects were predicted for each compound considering the following possibilities: inducer, inhibitor, or substrate. All analyzed compounds were predicted to act mainly as substrates (Figure 3A). The Venn diagram shows the number of effects predicted for the parent compound and for each class of metabolites, as well as the common effects (Figure 3B). AOH was predicted to have the most interactions with cytochromes P450 (16), followed by metabolites from aromatic hydroxylation (15).

The lowest effect was predicted for the metabolites resulting from o-glucuronidation (3 predictions) and o-sulfation (4 predictions). All the analyzed compounds were predicted as CYP2A11 substrates with a probability between 73.9% and 86.3%, but this was the only effect in common. AOH, metabolites M1–M3 and M10–M12, had high probability to be substrates for Cyp1A1, Cyp1A2, and Cyp2C12 (with a probability > 92%), or for Cyp1A6, Cyp2B6, Cyp2C, and Cyp2C9 (with a probability > 70%). AOH, as well as the metabolites resulting from aromatic hydroxylation and methylation, had a high probability prediction to be inhibitors for ubiquinol-cytochrome-c reductase (>80%). The prediction effects of parent mycotoxins and predicted metabolites on different isoforms of Phase II enzymes (Uridine 5′-diphospho-glucuronosyltransferase-UGT, glutathione S-transferase-GST, sulphotransferase-SULT) involved in xenobiotic metabolization are presented in Figure 4.

Similar to the effects predicted for cytochrome P450, all analyzed compounds were predicted to act mainly as substrates (Figure 4A). The Venn diagram shows the number of effects predicted for all the analyzed compounds (Figure 4B). Similar to the cytochrome P450 prediction, AOH and metabolites from aromatic hydroxylation had the most interactions with the enzymes of Phase II involved in toxin metabolization (7 for AOH and 5 for M1–M3). There was no common prediction for all the analyzed compounds. The lowest effect was predicted for the metabolites resulting from o-sulfation (4 predictions). AOH and metabolites from aromatic hydroxylation and methylation had the highest probability of acting as substrates for UGT1A6 (probability > 91%). Metabolites from o-glucuronidation were predicted to act as substrates for UGT1A7 (probability > 93%), while metabolites from o-sulfation act as substrates for GST and SULT (probability > 94%). Furthermore, PASS online software was used to predict the involvement of the analyzed compounds in oxidative stress (Figure 5).

Metabolites resulting from aromatic hydroxylation had the highest predicted probability involvement in the reactions associated with the oxidative stress (12 predicted reactions), followed by AOH (10 predicted reactions), as presented in the Venn diagram (Figure 5B). The lowest involvement in oxidative stress was predicted for the metabolites resulting from the reaction of o-sulfation with only two predicted reactions. All the analyzed compounds were predicted to inhibit the expression of HIF1A (>73%). With the exception of metabolites M7–M9, all the compounds were predicted to enhance TP53 expression and inhibit lipid peroxidase (>71%). Only AOH and metabolites M1–M3 may enhance the expression of heme-oxygenase 1 (HMOX1) (probability > 73%). AOH and metabolites from aromatic hydroxylation and methylation were predicted to inhibit some enzymes involved in the oxidative stress response, such as histidine kinase, nitrate reductase, peroxidase, and aldehyde oxidase (probability > 71%). Metabolites M1–M3 and M10–M12 were predicted to inhibit Jak2 expression (probability > 72%). As already described in Figure 5, only the metabolites resulting from the o-sulfation reaction were predicted to act as substrates for GST.

## 3. Discussion

According to the last scientific report of the EFSA Panel on Contaminants in the Food Chain, the estimated chronic dietary exposure to *Alternaria* toxins exceeds the threshold of toxicological concern [9].

Taking into consideration this aspect, together with insufficient relevant metabolism and toxicity data on *Alternaria* toxins that does not allow a proper health risk assessment for animal and public health, EFSA has indicated a need for additional compound-specific toxicity data [9].

The present study aimed to predict Phase I and II metabolites of alternariol as well as develop a detailed ADME/Tox profile for AOH and predicted metabolites using an in-silico workflow based on MetaTox, SwissADME, pKCMS, and PASS online software of computational toxicology in order to provide additional toxicological data for AOH toxicity.

MetaTox software was used to predict the metabolites of AOH that resulted from Phase I and II reactions in order to describe the metabolomic profile of AOH [33], for which twelve metabolites were predicted (Figure 1). AOH was predicted to generate three metabolites for the Phase I reaction (M1–M3) from aromatic hydroxylation. For the Phase II reaction, nine metabolites were predicted to result from the reaction of o-glucuronidation (M4–M6), o-sulfation (M7–M9), and methylation (M10–M12). The existence of modified forms of AOH have been reported in publications over the last few years [34]. For example, AOH sulfate conjugates can be produced by the *Alternaria alternata* fungus, whereas plant tissue can convert alternariol and alternariol-9-O-methyl ether into glucosylated metabolites [35].

The formation of glucuronides and sulfates of AOH and AME was reported as metabolites of AOH and alternariol monomethyl ether in hepatic and extrahepatic microsomes of rats, pigs, and humans, as well as in cultured human Caco-2 cells [36,37]. However, these sporadic data cannot provide a metabolomic profile like that predicted by using the MetaTox software, which uses an integrated assessment of the biotransformation reaction probabilities and their sites by utilizing the algorithm of PASS [33]. Additionally, the effects of these metabolites are unknown, and our study reports for the first time the prediction of physicochemical properties, pharmacokinetic predictions, drug likeness, and toxic effects related to the metabolomics profile of AOH.

With the exception of metabolites M4–M6, AOH and the predicted metabolites M1–M3 and M7–M12 have a moderate or high probability of being absorbed in the gut. These results are in agreement with the assessment of drug likeness via SwissADME, showing that the metabolites that resulted from the reaction of O-glucuronidation (M4–M6) violate the Lipinski rule of five for oral availability (Table 2).

Previous studies using cultured Caco-2 cells as a widely accepted in vitro system to evaluate human intestinal absorption and metabolism of drugs have shown that AOH appeared to be faster absorbed than AME from the gastrointestinal tract and may reach the portal blood both as aglycone and glucuronide and sulfate conjugate [37]. Another recent study has also shown that phase II metabolites of AOH and AME isolated from plants are absorbed and further metabolized by Caco-2 cells, which implicates their possible in vivo absorption in the animal gut [38]. However, it was predicted that none of these compounds can penetrate the blood–brain barrier or CNS (Table 3).

There is strong evidence concerning the genotoxic effects of AOH, which was reported to be mutagenic in in vitro tests as it can induce DNA strand break and act as a topoisomerases poison [39]. Additionally, *Alternaria* toxins were described as having other toxic effects, such as cytotoxic, mutagenic, carcinogenic, and endocrine disruptors [4]. In addition to these toxic effects, the computational model predicted other toxicological endpoints, such as vascular toxicity, hematotoxicity, diarrhea, and nephrotoxicity, for which in vivo evidence does not exist. As shown in the heatmap, several toxic effects (Figure 2) are common for the analyzed compounds, suggesting they can have a potential contribution to the overall toxicity of *Alternaria* toxins, as drugs with common modes of action may act jointly to produce higher combination effects than those of each single drug (Scher, 2012). For this reason, the combinatory effects of the parent toxin and its modified forms should be considered in the health risk assessment study of mycotoxins [40].

Cytochrome P450 (CYP450) are a family of enzymes with a key role in the metabolism of drugs and other xenobiotics, including mycotoxins [41,42]. Cyp450 enzymes can convert drugs/xenobiotics into less toxic compounds in order to prevent their toxicity or, on the contrary, generate reactive products (metabolites) that can cause toxicity [43].

AOH have been reported to enhance the expression of CYP1A cytochrome in human cells of different tissue origin, especially in esophageal cells [44,45]. Another recent in vitro study has shown that AOH is a potent inhibitor of CYP1A2, CYP2C9, but not of CYP2C19, 2D6, and 3A4 [46]. The computational model predicted that AOH had high probability to be substrate for Cyp1A1, Cyp1A2 but also for Cyp2C12, Cyp1A6, Cyp2B6, Cyp2C, and Cyp2C9 (Figure 3). These effects were also predicted for the metabolites M–M3 and M10–M12, while all the analyzed compounds may be a substrate for CYP2A11.

Phase II drug-metabolizing enzymes involved in conjugation are mainly transferases with important roles in the conjugation of xenobiotics or endogenous compounds and the transformation of the parent compound into inactive metabolites, which can be easily excreted by urine [47]. Previous studies have shown that glucuronidation realized by UGT represents a major metabolic pathway for AOH transformation in hepatic and extrahepatic tissues, and that at least nine human UGTs (1A1, 1A3, 1A6, 1A7, 1A9, 1A10, 2B7, and 2B15) were able to conjugate AOH [36]. Similarly, our data have shown that AOH was predicted to act as a substrate for five UGTs (1A1, 1A3, 1A6, 1A8, and 12A9). As shown in Figure 4, metabolites from aromatic hydroxylation, methylation, o-glucuronidation, and o-sulfation were also predicted as substrates for UGT1As, while only metabolites from o-sulfation were predicted as substrates for UGT2Bs.

Other phase II enzymes are sulfotransferases catalyzing the sulfation of xenobiotics or endogenous compounds [48]. AOH-sulfates that resulted from the sulfation reaction were detected in plants [14,49,50] and as compounds resulting from AOH metabolism in a CaCo2 cell model [37], or in the urine and feces of rats [51]. With the exception of the metabolites that resulted from methylation, all the other compounds were predicted to be substrates for sulfotransferase (Figure 4).

Glutathione S-transferases are Phase II detoxification enzymes that catalyze the glutathione conjugation, playing an important role in the cellular detoxification system and protecting cells from oxidative stress [52]. Previous studies have shown that alternariol induces oxidative stress, DNA damage, and cell cycle arrest in cells of different origins [16] but the mechanisms involved are not clearly understood. AOH and 4-OH-AOH were shown to increase reactive oxygen species (ROS) synthesis [16], leading to an increase in lipid peroxidation (LPO) [53].

The analysis of the interaction of AOH with different markers of oxidative stress (Figure 6) has shown a high probability of similar interactions, at least between AOH and the metabolites of aromatic hydroxylation M1–M3 (9 common predictions), between M1–M3 and the metabolites of methylation M10–M12 (9 common predictions), or between AOH and M10–M12 (7 common predictions). Inhibition of transcription factor hypoxia inducible factor-1α (HIF1A) expression, which mediates adaptive responses to oxidative stress [54], was predicted as a common effect for all analyzed compounds, suggesting the important role of the HIF1A-1α pathway in controlling oxidative stress.

## 4. Conclusions

In silico approaches represent a useful tool for estimating the toxicity of mycotoxins in order to predict toxicity, provide some preliminary information concerning the toxic effect of the tested compound(s), and offer guidance for in vitro and in vivo toxicity tests. As predicted by MetaTox software, 12 metabolites were identified as corresponding to the metabolomic profile of alternariol. Our study reports for the first time the prediction of physicochemical properties, pharmacokinetic predictions, drug likeness, and toxic effects related to the metabolomic profile of AOH. The ADME profile for AOH and predicted metabolites indicated a moderate or high intestinal absorption probability for AOH and all metabolites, except for the metabolites that resulted from O-glucuronidation; however, a low probability of penetration of the blood–brain barrier was demonstrated. As shown in our data, the metabolites that resulted from the aromatic hydroxylation reaction have similar toxicological endpoints to the parent compound, while the metabolites that resulted from glucuronidation and sulfation show a wide and different toxicity profile.

Besides the cytotoxic, mutagenic, carcinogenic, and endocrine disruptor effects, the computational model predicted other toxicological endpoints, such as vascular toxicity, haemato-toxicity, diarrhea, and nephrotoxicity. AOH and its metabolites have been predicted to act as substrates for different isoforms of phase I and II drug-metabolizing enzymes and to interact with the response to oxidative stress. Among all analyzed compounds, AOH and the metabolites that resulted from aromatic hydroxylation were predicted to have the most interactions with the enzymes of Phase I and Phase II. Oxidative stress was predicted to play an important role in AOH and its metabolites toxicity and inhibition of transcription factor hypoxia inducible factor-1α (HIF1A) expression, which mediates adaptive responses to oxidative stress and was predicted as a common effect for all analyzed compounds, suggesting the important role of the HIF1A-1α pathway in the oxidative stress induced by AOH and its metabolites.

All these data concerning toxicity prediction may indicate, besides the individual toxicity of AOH and its metabolites, a possible increasing effect on overall AOH toxicity as the compounds with common modes of action may act jointly to produce higher combination effects than those of each single drug. However, these in silico approaches have some limitations, and all the predicted results should be confirmed in the future by in vitro and in vivo toxicity tests that may validate the metabolic transformation of AOH in the predicted metabolites as well as their ADME/Tox profile.

## 5. Materials and Methods

### 5.1. Prediction of Alternariol Metabolites

MetaTox is a free web application (http://www.way2drug.com/mg, accessed on 22 August 2022) that integrates metabolism pathway generation with the prediction of acute toxicity. MetaTox uses the Marvin JS chemical editor to input and visualize the molecular structure [33].

For metabolite prediction, the SMILE (simplified molecular-input line-entry system) canonical structure for AOH was submitted to the online server MetaTox in order to predict the xenobiotic metabolism and the possible sites of metabolism from biotransformation reactions of phase I and II. MetaTox predicts the metabolomic profile resulting from nine classes of reactions (aliphatic and aromatic hydroxylation, N- and O-glucuronidation, N-, S-, and C-oxidation, and N- and O-dealkylation) catalyzed by human enzymes of Phase I and II of drug metabolism (cytochromes P450s and UDP glucuronysil transferases) [33]. The calculation of probability for the generated metabolites is based on analyses of “structure-biotransformation reactions” and “structure-modified atoms” relationships using a Bayesian approach. A cut-off value of 0.9, “no-limit” for metabolite-likeness, and a layer count value of 1 were chosen as the parameters for the prediction of the metabolites for the AOH structure.

### 5.2. Prediction of Physicochemical Properties, Pharmacokinetic Predictions and Drug Likeness

Physically relevant properties and pharmaceutically-relevant descriptors of the alternariol and metabolites were then predicted using Swiss ADME (http://www.swissadme.ch, accessed on 23 September 2022) and pKCMS (https://biosig.lab.uq.edu.au/pkcsm, accessed on 26 September 2022) free online software. The physicochemical properties (number of heavy atoms, number of aromatic heavy atoms, number of rotatable bonds, number of hydrogen bond donors/number of hydrogen bond acceptors, topological polar surface area), lipophilicity (cLog P_o/w_), and water solubility were predicted using SwissADME, a free web tool to evaluate pharmacokinetics, drug-likeness, and the medicinal chemistry friendliness of small molecules [23]. Drug likeness was performed via SwissADME in order to evaluate the molecular properties of AOH and the predicted metabolites in the human body based on Lipinski’s rule of five.

Pharmacokinetic properties related to absorption (water solubility, CaCo_2_ permeability, intestinal absorption, skin permeability, P-glycoprotein substrate), distribution (blood–brain barrier (BBB) and central nervous system (CNS) permeability, volume distribution (VD), fraction unbound (Fu)) metabolism (substrate or inhibitors for P450 cytochromes), and excretion (total clearance and renal OCT2 substrate), which are essential parameters for the prediction of the drugs ADME [55], were predicted using pKCMS, a software that predicts small-molecule pharmacokinetic and toxicity properties using graph-based signatures [24].

### 5.3. Prediction of Toxicity

The toxicological endpoints (Genotoxic, Toxic vascular, Hematemesis, Hematotoxic, Carcinogenic, Teratogen, Embryotoxic, Endocrine disruptor, Inflammation, Mutagenic, Hypercholesterolemic, Nephrotoxic, Reproductive dysfunction, Diarrhea) were predicted using PASS Prediction of Activity Spectra for Substances (PASS) *(*http://www.way2drug.com/passonline, accessed on 6 February 2023). PASS is an online free software that allows the evaluation of the biological activity profile of an organic compound based on its chemical structure. PASS can provide simultaneous predictions of different types of biological activity with two probabilities: the probability “to be active” (Pa) or the probability “to be inactive” (Pi), expressed as a percentage of the probability. SMILE specifications of AOH and predicted metabolites M1–M12 were submitted to PASS online software in order to predict, in addition to their toxicological endpoints, the interaction with different isoforms of Phase I and Phase II enzymes involved in the xenobiotic metabolization or their involvement in the reactions related to oxidative stress. Toxicity profiles against different organisms: *Salmonella* (AMES toxicity), *Tetrahymena pyriformis* (*T. pyriformis* toxicity), minnow (Minnow toxicity), rats (oral acute toxicity—LD50, oral chronic toxicity—LOAEL), and humans (maximum tolerated dose) were predicted using pKCMS software. The overall procedures followed the in silico approach. The metabolomic and toxicity predictions for alternariol are presented in Figure 6.

## Figures and Tables

**Figure 1 toxins-15-00421-f001:**
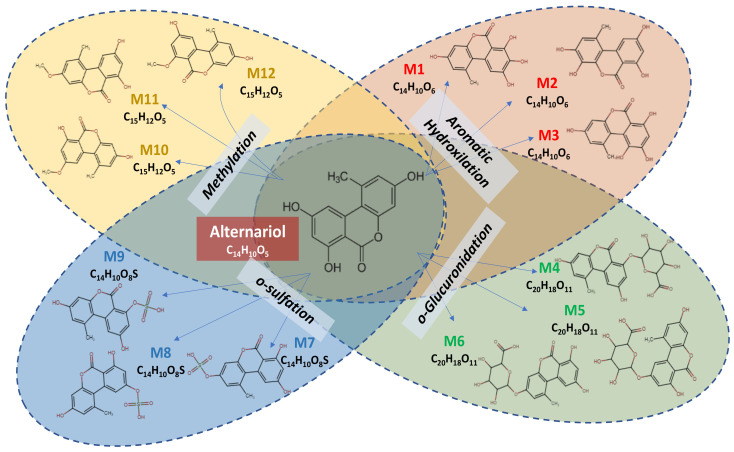
Phase I and II metabolites of alternariol predicted by MetaTox software.

**Figure 2 toxins-15-00421-f002:**
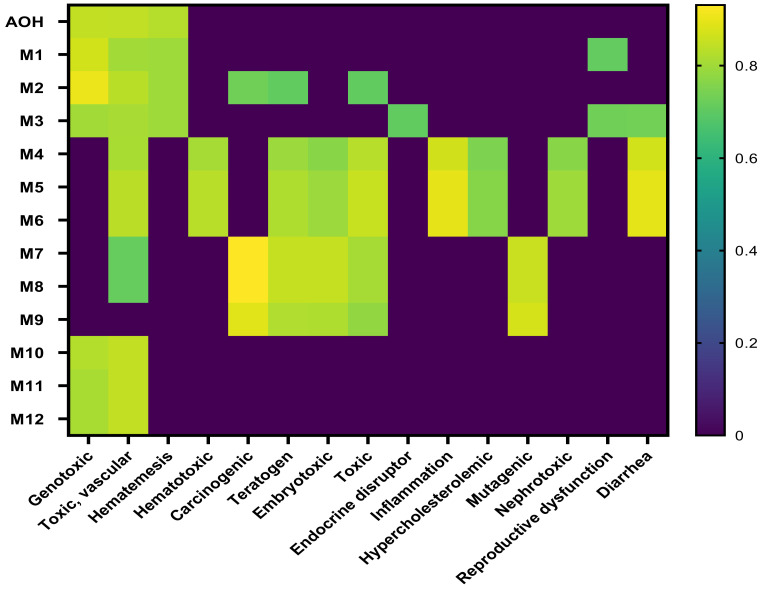
Heatmap of predicted toxic effects of AOH and its metabolites.

**Figure 3 toxins-15-00421-f003:**
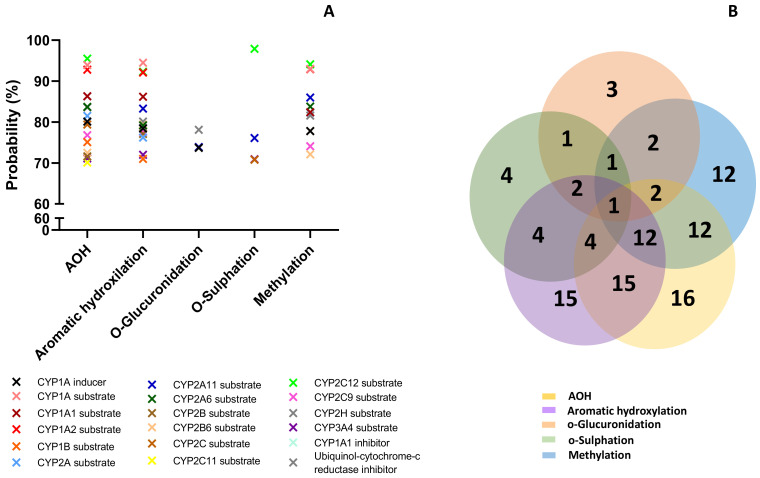
Prediction effects of AOH and predicted metabolites on different isoforms of Phase I enzymes (**A**). Venn diagram showing the number of effects predicted for the parent compound and for each class of metabolites (**B**).

**Figure 4 toxins-15-00421-f004:**
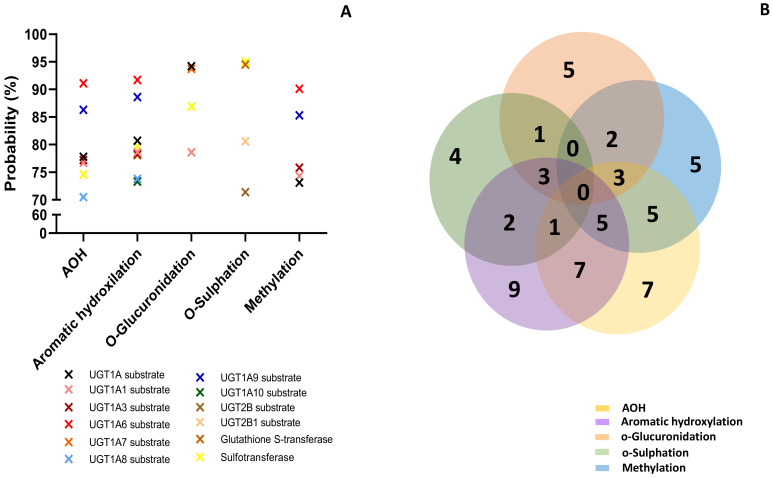
Prediction effects of AOH and predicted metabolites on different isoforms of Phase II enzymes (**A**). Venn diagram showing the number of effects predicted for the parent compound and for each class of metabolites (**B**).

**Figure 5 toxins-15-00421-f005:**
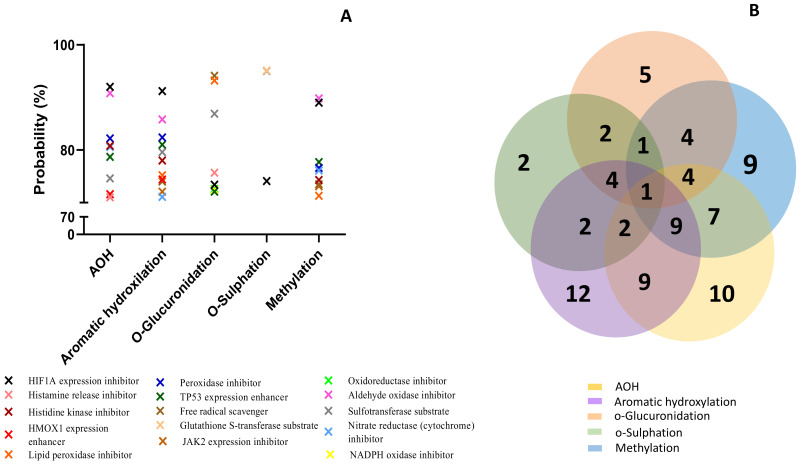
Prediction effects of AOH and predicted metabolites on different isoforms of Phase II enzymes (**A**). Venn diagram showing the number of effects predicted for the parent compound and for each class of metabolites (**B**).

**Figure 6 toxins-15-00421-f006:**
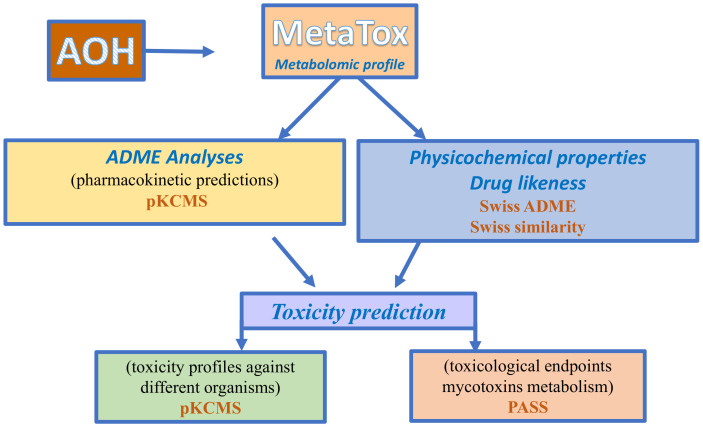
Workflow used for the in-silico approach for metabolomic and toxicity prediction for alternariol.

**Table 1 toxins-15-00421-t001:** Canonical SMILES for alternariol and its MetaTox predicted metabolites.

Parent Compound/Metabolite	Canonical SMILES
Alternariol (AOH)	CC1=CC(O)=CC2=C1C1=CC(O)=CC(O)=C1C(=O)O2
Aromatic Hydroxylation	
Metabolite (M1)	CC1=CC(O)=CC2=C1C1=CC(O)=C(O)C(O)=C1C(=O)O2
Metabolite (M2)	CC1=CC(O)=C(O)C2=C1C1=CC(O)=CC(O)=C1C(=O)O2
Metabolite (M3)	CC1=CC(O)=CC2=C1C1=C(O)C(O)=CC(O)=C1C(=O)O2
O-Glucuronidation	
Metabolite (M4)	CC1=CC(O)=CC2=C1C1=CC(O)=CC(OC3OC(C(O)C(O)C3O)C(O)=O)=C1C(=O)O2
Metabolite (M5)	CC1=CC(O)=CC2=C1C1=CC(OC3OC(C(O)C(O)C3O)C(O)=O)=CC(O)=C1C(=O)O2
Metabolite (M6)	CC1=CC(OC2OC(C(O)C(O)C2O)C(O)=O)=CC2=C1C1=CC(O)=CC(O)=C1C(=O)O2
O-Sulfation	
Metabolite (M7)	CC1=CC(OS(O)(=O)=O)=CC2=C1C1=CC(O)=CC(O)=C1C(=O)O2
Metabolite (M8)	CC1=CC(O)=CC2=C1C1=CC(OS(O)(=O)=O)=CC(O)=C1C(=O)O2
Metabolite (M9)	CC1=CC(O)=CC2=C1C1=CC(O)=CC(OS(O)(=O)=O)=C1C(=O)O2
Methylation	
Metabolite (M10)	COC1=CC(O)=C2C(=O)OC3=CC(O)=CC(C)=C3C2=C1
Metabolite (M11)	COC1=CC(C)=C2C(OC(=O)C3=C(O)C=C(O)C=C23)=C1
Metabolite (M12)	COC1=CC(O)=CC2=C1C(=O)OC1=CC(O)=CC(C)=C21

**Table 2 toxins-15-00421-t002:** Lipinski’s molecular descriptors for alternariol and its MetaTox predicted metabolites from SwissADME.

Parent Compound/Metabolite	MW (g/mol) (≤500)	HBA (≤10)	HBD (≤5)	cLogP (<5)	MR (40–130)	n-ROTB	TPSA (Å^2^)
Alternariol (AOH)	258.23	5	3		71.03	0	90.9
Aromatic Hydroxylation							
Metabolite (M1)	274.23	6	4	1.71	73.05	0	111.1
Metabolite (M2)				1.72			
Metabolite (M3)				1.46			
O-Glucuronidation							
Metabolite (M4)	434.35	11 *	6 *	0.06	103.79	3	187.1
Metabolite (M5)				0.31			
Metabolite (M6)				0.31			
O-Sulfation							
Metabolite (M7)	338.29	8	3	1.58	81.22	3	142.6
Metabolite (M8)				1.57			
Metabolite (M9)				1.38			
Methylation							
Metabolite (M10)	272.25	5	2	2.55	75.49	1	79.9
Metabolite (M11)				2.55			
Metabolite (M12)				2.55			

MW—molecular weight; HBD—Hydrogen bond donor; HBA = Hydrogen bond acceptor; cLogP—lipophilicity; MR—molar refractivity; n-ROTB: number of rotatable bounds; TPSA = Topological polar surface area; * Denotes violation of Lipinski’s rule of five.

**Table 3 toxins-15-00421-t003:** ADME prediction for AOH and metabolites.

Model Name/Parameters	AOH	Aromatic Hydroxilation			O-Glucuronidation			O-Sulfation			Methylation		
		M1	M2	M3	M4	M5	M6	M7	M8	M9	M10	M11	M12
**Absorption**													
Water solubility (log mol/L)	−2.982	−3.059	−3.151	−3.072	−2.892	−2.894	−2.894	−3.158	−3.118	−2.938	−3.5	−3.293	−3.388
Caco2 permeability (log Paap in 10^−6^ cm/s)	1.025	0.838	0.815	0.818	−0.745	−0.699	−0.885	0.388	0.635	0.717	1.057	0.952	0.9
Intestinal absorption (human) %	95.473	73.662	81.84	76.718	18.049	18.043	14.941	45.397	48.67	47.87	95.627	95.804	97.087
Skin Permeability (log Kp)	−2.745	−2.735	−2.735	−2.735	−2.735	−2.735	−2.735	−2.735	−2.735	−2.735	−2.739	−2.737	−2.747
P-glycoprotein substrate	Yes	Yes	Yes	Yes	Yes	Yes	Yes	Yes	Yes	Yes	Yes	Yes	Yes
P-glycoprotein I inhibitor	No	No	No	No	No	No	No	No	No	No	No	No	No
P-glycoprotein II inhibitor	No	No	No	No	No	No	No	No	No	No	No	No	No
**Distribution**													
VDss (human) (log L/kg)	−0.032	0.064	0.144	0.215	−0.891	−0.928	−1.032	−0.557	−0.306	−0.402	−0.009	−0.061	0.208
Fraction unbound (human) (Fu)	0.14	0.083	0.11	0.112	0.22	0.202	0.193	0.163	0.176	0.223	0.125	0.164	0.164
BBB permeability (log BB)	−0.965	−1.32	−1.218	−1.325	−1.736	−1.647	−1.834	−1.556	−1.397	−1.48	−0.107	−0.225	−0.16
CNS permeability (log PS)	−2.247	−2.557	−2.499	−2.498	−4.452	−4.438	−4.551	−3.76	−3.565	−3.5	−2.236	−2.24	−2.218
**Metabolism**													
CYP2D6 substrate	No	No	No	No	No	No	No	No	No	No	No	No	No
CYP3A4 substrate	No	No	No	No	No	No	No	No	No	No	No	No	No
CYP1A2 inhibitor	Yes	Yes	Yes	Yes	No	No	No	Yes	Yes	No	Yes	Yes	Yes
CYP2C19 inhibitor	No	No	No	No	No	No	No	No	No	No	Yes	Yes	Yes
CYP2C9 inhibitor	Yes	Yes	No	No	No	No	No	No	No	No	No	Yes	Yes
CYP2D6 inhibitor	No	No	No	No	No	No	No	No	No	No	No	No	No
CYP3A4 inhibitor	No	No	No	No	No	No	No	No	No	No	No	No	No
**Excretion**													
Total Clearance (log mL/min/kg)	0.723	0.658	0.676	0.717	0.77	0.785	0.815	0.841	0.848	0.831	0.841	0.82	0.79
Renal OCT2 substrate	No	No	No	No	No	No	No	No	No	No	No	No	No

**Table 4 toxins-15-00421-t004:** Numeric and categorical units for toxicity for AOH and predicted metabolites.

Toxicity	AOH	Reaction/Metabolites											
		Aromatic Hydroxilation			O-Glucuronidation			O-Sulfation			Methylation		
		M1	M2	M3	M4	M5	M6	M7	M8	M9	M10	M11	M12
AMES toxicity	Yes	Yes	No	Yes	No	No	No	No	No	No	No	Yes	No
Max. tolerated dose (human)	0.697	0.94	0.832	0.888	0.681	0.693	0.654	0.976	0.945	0.918	0.489	0.765	0.472
Oral Rat Acute Toxicity (LD50)	2.82	2.611	2.515	2.586	2.438	2.441	2.466	2.582	2.63	2.509	2.918	3.001	3.036
Oral Rat Chronic Toxicity (LOAEL)	1.51	2.747	2.675	2.625	4.409	4.101	3.945	1.917	2.169	2.158	1.285	0.98	0.944
*T. pyriformis* toxicity	0.335	0.301	0.305	0.297	0.285	0.285	0.285	0.285	0.285	0.285	0.339	0.349	0.348
Minnow toxicity	1.512	1.235	1.54	2.128	3.918	3.476	4.205	0.414	1.586	1.927	0.332	−0.352	0.319

## Data Availability

Data can be available on request.

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
