# Peer review of "Using In Silico Approach for Metabolomic and Toxicity Prediction of Alternariol"

_toxins, 2023, doi:10.3390/toxins15070421_

Round 1
Reviewer 1 Report
1. The authors should add the detailed note on the Phase I and II metabolites present in various foods and its adverse effects on human/animal health.
2. Also, future perspectives from the current research outcomes should be explained in the conclusion section.
3. Methodology section needs to explain clearly; block diagram is preferable.
Author Response
First of all the Authors want to thanks to the Reviewer for the very useful comments and observations that will contribute to the substantial increase of the manuscript quality.
Reviewer comment: The authors should add the detailed note on the Phase I and II metabolites present in various foods and its adverse effects on human/animal health.
Author response: Thank you for your comment. As response to your suggestion, we have included in the Introduction section information related to the Phase I and II metabolites present in various foods and its adverse effects on human/animal health.
“Few data concerning the toxicological effects of AOH metabolites are available in the literature. For example, DNA strand breaking potential of AOH decrease significantly following hydroxylation or glucuronidation reaction (Tiessen et al., 2017; Pfeiffer et al., 2007). Also, AOH and its metabolite, AOH-3-O-sulphate may have a similar interaction with estrogen receptor as described in a silico study performed by Dellafiora et al., 2017, that indicate a possible comparative estrogenic effect of the two compounds.”
Reviewer comment: Also, future perspectives from the current research outcomes should be explained in the conclusion section.
Author response: Thank you for pointing out this regrettable omission from our Conclusion. According to this recommendation together with other suggestion and comments of the other Reviewers and Academic Editor we have now reformulated the Conclusion section in the new manuscript version:
“In-silico approaches represent a useful tool for estimating the toxicity of mycotoxins in order to predict toxicity, that can provide some preliminary information concerning the toxic effect of the tested compound(s) and could offer a guidance for the in vitro and in vivo toxicity tests. As predicted by MetaTox software, a number of 12 metabolites were identified as corresponding to the metabolomic profile of alternariol. Our study reports for the first time the prediction of physicochemical properties, pharmacokinetic predictions, drug likeness and toxic effects related to the metabolomics profile of AOH. ADME profile for AOH and predicted metabolites indicated a moderate or high intestinal absorption probability for AOH and all metabolites excepting the metabolites resulted from O-glucuronidation, but a low probability to penetrate the blood brain barrier. As resulted from our data, the metabolites resulted from aromatic hydroxylation reaction have similar toxicological endpoints as the parent compound, while the metabolites resulted from glucuronidation and sulphation show a wide and different toxicity profile. Beside cytotoxic, mutagenic, carcinogenic, endocrine disruptor effects, the computational model has predicted for AOH and predicted metabolites other toxicological endpoints as vascular toxicity, hematotoxicity, diarrhea, nephrotoxicity. AOH and its metabolites have been predicted to act as substrate for different isoforms of phase I and II drug-metabolizing enzymes and to interact with the response to oxidative stress. Among all analyzed compounds, AOH and the metabolites resulted from aromatic hydroxylation were predicted to have the most interactions with the enzymes of Phase I and Phase II. Oxidative stress was predicted to have an important role in AOH and its metabolites toxicity and inhibition of transcription factor hypoxia inducible factor-1α (HIF1A) expression that mediates adaptive responses to oxidative stress was predicted as a common effect for all analyzed compounds, suggesting an important role of HIF1A-1α pathway in the oxidative stress induced by AOH and its metabolites.
All these data concerning toxicity prediction may indicate beside the individual toxicity of AOH and its metabolites, a possible increase effect on the overall AOH toxicity, as the compounds with common modes of action may act jointly to produce higher combination effects that the effects of each single drug. However, these in silico approaches have some limitations and all the predicted results should be confirmed in the future by in vitro and in vivo toxicity tests that may validate the metabolic transformation of AOH in the predicted metabolites as well as their ADME/Tox profile.”
Reviewer comment: Methodology section needs to explain clearly; block diagram is preferable.
Author response. Thank you for your comment. We have replaced the figure 6 that illustrate the workflow used for the in-silico approach for metabolomic and toxicity prediction for alternariol with a block diagram in order to facilitate the understanding of the used methodology.
Reviewer 2 Report
The article titled “Using in silico approach for metabolomic and toxicity prediction of alternariol”. The authors in the present study did a comprehensive in silico-based investigation to find out how the common fungal toxin alternariol is metabolized inside the human body. The authors found out that alternariol can probably be converted into 12 different metabolites. Subsequently, these putative metabolites were predicted for their toxicity and possible pharmacological activities. Accordingly, the authors assumed that such in silico methods can represent a viable alternative to the in vitro and in vivo tests for the prediction of mycotoxins metabolism and toxicity.
I totally do not agree with the authors in this assumption and see that despite their comprehensive work, the final conclusions and findings are just predictions that are now considered as routine rapid, and simple preliminary work that has to be validated by subsequent in vitro and in vivo experiments.
Hence, I don’t find that this paper in its current form is suitable for publication as it did not provide the readers with any solid new information.
The English quality is average, and may need proof reading.
Author Response
First of all, the Authors want to thanks to the Reviewer for the very useful comments and observations that will contribute to the substantial increase of the manuscript quality.
Reviewer comment: The article titled “Using in silico approach for metabolomic and toxicity prediction of alternariol”. The authors in the present study did a comprehensive in silico-based investigation to find out how the common fungal toxin alternariol is metabolized inside the human body. The authors found out that alternariol can probably be converted into 12 different metabolites. Subsequently, these putative metabolites were predicted for their toxicity and possible pharmacological activities. Accordingly, the authors assumed that such in silico methods can represent a viable alternative to the in vitro and in vivo tests for the prediction of mycotoxins metabolism and toxicity.
Reviewer comment: I totally do not agree with the authors in this assumption and see that despite their comprehensive work, the final conclusions and findings are just predictions that are now considered as routine rapid, and simple preliminary work that has to be validated by subsequent in vitro and in vivo experiments.
Reviewer comment: Hence, I don’t find that this paper in its current form is suitable for publication as it did not provide the readers with any solid new information.
Authors response: The Authors want to thanks the Reviewer for the comments and appreciations of the present paper. The in-silico approaches represent a useful tool for estimating the toxicity of mycotoxins in order to predict toxicity, that can provide some preliminary information concerning the toxic effect of the tested compound(s) and offering a guidance for the in vitro and in vivo toxicity tests. To the best of our knowledge, beside the possible xenoestrogenic effects, the toxic effects of these metabolites are unknown and our study reports for the first time the prediction of physicochemical properties, pharmacokinetic predictions, drug likeness and toxic effects related to the metabolomics profile of AOH. We totally agree with the reviewer that these in silico approaches have some limitations and all the predicted results should be confirmed in the future by in vitro and in vivo toxicity tests. In consequence, as suggested we have modified to Conclusion section and we have included this work perspective in the new version of the manuscript:
“In-silico approaches represent a useful tool for estimating the toxicity of mycotoxins in order to predict toxicity, that can provide some preliminary information concerning the toxic effect of the tested compound(s) and could offer a guidance for the in vitro and in vivo toxicity tests. As predicted by MetaTox software, a number of 12 metabolites were identified as corresponding to the metabolomic profile of alternariol. Our study reports for the first time the prediction of physicochemical properties, pharmacokinetic predictions, drug likeness and toxic effects related to the metabolomics profile of AOH. ADME profile for AOH and predicted metabolites indicated a moderate or high intestinal absorption probability for AOH and all metabolites excepting the metabolites resulted from O-glucuronidation, but a low probability to penetrate the blood brain barrier. As resulted from our data, the metabolites resulted from aromatic hydroxylation reaction have similar toxicological endpoints as the parent compound, while the metabolites resulted from glucuronidation and sulphation show a wide and different toxicity profile. Beside cytotoxic, mutagenic, carcinogenic, endocrine disruptor effects, the computational model has predicted for AOH and predicted metabolites other toxicological endpoints as vascular toxicity, hematotoxicity, diarrhea, nephrotoxicity. AOH and its metabolites have been predicted to act as substrate for different isoforms of phase I and II drug-metabolizing enzymes and to interact with the response to oxidative stress. Among all analyzed compounds, AOH and the metabolites resulted from aromatic hydroxylation were predicted to have the most interactions with the enzymes of Phase I and Phase II. Oxidative stress was predicted to have an important role in AOH and its metabolites toxicity and inhibition of transcription factor hypoxia inducible factor-1α (HIF1A) expression that mediates adaptive responses to oxidative stress was predicted as a common effect for all analyzed compounds, suggesting an important role of HIF1A-1α pathway in the oxidative stress induced by AOH and its metabolites.
All these data concerning toxicity prediction may indicate beside the individual toxicity of AOH and its metabolites, a possible increase effect on the overall AOH toxicity, as the compounds with common modes of action may act jointly to produce higher combination effects that the effects of each single drug. However, these in silico approaches have some limitations and all the predicted results should be confirmed in the future by in vitro and in vivo toxicity tests that may validate the metabolic transformation of AOH in the predicted metabolites as well as their ADME/Tox profile.”
Reviewer 3 Report
The occurrence and toxic effects of Alternaria toxins, especially AOH, and their “masked mycotoxins” have been increasing evidenced. Even there are no regulations at present concerning AOH guidance concentration in food and feed, the human dietary exposure to AOH exceeds the threshold of toxicological concern, and an additional compound-specific toxicity data should be needed according to EFSA. Thus ,this article deals with an especially significant problem, and the author gives an useful working model based on online computational programs to predict AOH and its metabolites.
I have the following note:
1. "Alternaria" should be in italics
2. In the Introduction section, the previous research and results on the in silico working model were lacked, which should be added.
3. There are two Table 3 in this paper, and the titles are completely different.
4. In the first Table 3, AOH should be added in the Model name/parameters.
5. The accuracy and validity of this model need to be verified if possible.
Author Response
The Authors want to thanks to the Reviewer for the very useful comments and observations that will contribute to the substantial increase of the manuscript quality.
Comments and Suggestions for Authors
The occurrence and toxic effects of Alternaria toxins, especially AOH, and their “masked mycotoxins” have been increasing evidenced. Even there are no regulations at present concerning AOH guidance concentration in food and feed, the human dietary exposure to AOH exceeds the threshold of toxicological concern, and an additional compound-specific toxicity data should be needed according to EFSA. Thus, this article deals with an especially significant problem, and the author gives a useful working model based on online computational programs to predict AOH and its metabolites.
Reviewer comment: "Alternaria" should be in italics
Author response. Thank you for this observation. In the new version of the manuscript the suggested correction was made.
Reviewer comment: In the Introduction section, the previous research and results on the in silico working model were lacked, which should be added.
Author response. Thank you for this valuable comment. Previous studies have used in silico approaches in order to assess the xenoestrogenic potential of Alternaria toxins in human and other species. Our study provides new data concerning the toxicity of AOH and its metabolites and reports for the first time the prediction of physicochemical properties, pharmacokinetic predictions, drug likeness and toxic effects related to the metabolomics profile of AOH. As suggested, information about the previous research and results on Alternaria toxins using the in silico model were now included in the new version of the manuscript:
“Previous studies have used in silico approaches in order to assess the estrogenic potential of Alternaria mycotoxins and other xenoestrogens. A recent study of Dellafiora et al, has shown that in silico approaches can be used as useful tools for assessing differences between species in terms of mycotoxin estrogenicity (Dellafiora et al., 2020). Additionally, in silico approach was used for assessing the xenoestrogenic potential of Alternaria mycotoxins and metabolites, indicating that methylation reaction can increase AOH estrogenicity (Dellafiora et al., 2018).”
Reviewer comment: There are two Table 3 in this paper, and the titles are completely different.
Author response: Thank you for observing this error from our part. Indeed, the table numbering was wrong and the requested correction was made in the new version of the manuscript.
Reviewer comment: In the first Table 3, AOH should be added in the Model name/parameters.
Author response: Thank you for this observation. We have made the suggested correction.
Reviewer comment: The accuracy and validity of this model need to be verified if possible.
Author response: The Authors are grateful to the Reviewer for this valuable comment. Indeed, the in silico methods offers only a prediction and the accuracy and validity of the model should be validated through in vitro and in vivo studies. This comment was now included in the new version of the manuscript:
“However, these in silico approaches have some limitations and all the predicted results should be confirmed in the future by in vitro and in vivo toxicity tests that may validate the metabolic transformation of AOH in the predicted metabolites as well as their ADME/Tox profile.”
Round 2
Reviewer 2 Report
I thank the authors for their response. Actually, I still do not see solid results (i.e., experimental) to support the predictions. Another issue to be considered by the author in their future work, each AI-based prediction software has a confidence and success rate that do not guarantee that all the retrieved results and scores will give positive outcomes in the lab. So, all the presented results in your paper are just preliminary speculations that do not provide the readers with solid results.
English needs some improvement.
Author Response
Thank you for your comments. As already mentioned in our manuscript, our study reports for the first time the prediction of physicochemical properties, pharmacokinetic predictions, drug likeness and toxic effects related to the metabolomics profile of AOH.
We agree that these are only predictions that can be or not confirmed , but as we have already mentioned these predictions should be further confirmed by the in vivo and in vitro studies.
As stated in the manuscript, the computational methods represent a viable alternative for identification of toxins harmful effect to the in vivo animal tests that are expensive, time consuming, rise ethical considerations, and should be limited according to principle of 3Rs. For this reason, in the last years there were a lot of peer reviewed publications that were based only on the prediction of the ADMET properties of different compounds and that have used only computational approaches
Please find bellow some examples from the literature:
Domínguez-Villa, F.X.; Durán-Iturbide, N.A.; Ávila-Zárraga, J.G. Synthesis, molecular docking, and in silico adme/tox profiling studies of new 1-aryl-5-(3-azidopropyl)indol-4-ones: Potential inhibitors of sars cov-2 main protease. Bioorganic chemistry 2021, 106, 104497.
Agahi, F.; Juan, C.; Font, G.; Juan-García, A. In silico methods for metabolomic and toxicity prediction of zearalenone, α-zearalenone and β-zearalenone. Food and Chemical Toxicology 2020, 146, 111818.
Ramírez, H., Fernandez-Moreira, E., Rodrigues, J.R. et al. Synthesis and in silico ADME/Tox profiling studies of heterocyclic hybrids based on chloroquine scaffolds with potential antimalarial activity. Parasitol Res 121, 441–451 (2022). https://doi.org/10.1007/s00436-021-07374-7
Han, Y., Zhang, J., Hu, C. Q., Zhang, X., Ma, B., & Zhang, P. (2019). In silico ADME and Toxicity Prediction of Ceftazidime and Its Impurities. Frontiers in pharmacology, 10, 434. https://doi.org/10.3389/fphar.2019.00434
Zadorozhnii, P.V.; Kiselev, V.V.; Kharchenko, A.V. In Silico ADME Profiling of Salubrinal and Its Analogues. Future Pharmacol. 2022, 2, 160-197. https://doi.org/10.3390/futurepharmacol2020013
Cheng F, Li W, Liu G, Tang Y. In silico ADMET prediction: recent advances, current challenges and future trends. Current Topics in Medicinal Chemistry. 2013 ;13(11):1273-1289. DOI: 10.2174/15680266113139990033. PMID: 23675935.
Wang, Y., Xing, J., Xu, Y., Zhou, N., Peng, J., Xiong, Z., . . . Jiang, H. (2015). In silico ADME/T modelling for rational drug design. Quarterly Reviews of Biophysics, 48(4), 488-515. doi:10.1017/S0033583515000190
El-Saadi MW, Williams-Hart T, Salvatore BA, Mahdavian E. Use of in-silico assays to characterize the ADMET profile and identify potential therapeutic targets of fusarochromanone, a novel anti-cancer agent. In Silico Pharmacol. 2015 Dec;3(1):6. doi: 10.1186/s40203-015-0010-5. Epub 2015 Jun 4. PMID: 26820891; PMCID: PMC4464579.
El fadili, M.; Er-Rajy, M.; Kara, M.; Assouguem, A.; Belhassan, A.; Alotaibi, A.; Mrabti, N.N.; Fidan, H.; Ullah, R.; Ercisli, S.; et al. QSAR, ADMET In Silico Pharmacokinetics, Molecular Docking and Molecular Dynamics Studies of Novel Bicyclo (Aryl Methyl) Benzamides as Potent GlyT1 Inhibitors for the Treatment of Schizophrenia. Pharmaceuticals 2022, 15, 670. https://doi.org/10.3390/ ph15060670
Reviewer 3 Report
-
The author has modified it according to my suggestion and I agree to accept it.
Author Response
The authors want to thanks to the Reviewer for the very useful comments and observations that contribute to the improvement of the manuscript quality.
Best regards,
The Authors